# OpenBloodFlow: A User-Friendly OpenCV-Based Software Package for Blood Flow Velocity and Blood Cell Count Measurement for Fish Embryos

**DOI:** 10.3390/biology11101471

**Published:** 2022-10-08

**Authors:** Ali Farhan, Ferry Saputra, Michael Edbert Suryanto, Fahad Humayun, Roi Martin B. Pajimna, Ross D. Vasquez, Marri Jmelou M. Roldan, Gilbert Audira, Hong-Thih Lai, Yu-Heng Lai, Chung-Der Hsiao

**Affiliations:** 1Department of Chemistry, Chung Yuan Christian University, Taoyuan 320314, Taiwan; 2Department of Bioscience Technology, Chung Yuan Christian University, Taoyuan 320314, Taiwan; 3Department of Bioinformatics and Biostatistics, Shanghai Jiao Tong University, Shanghai 200240, China; 4Faculty of Pharmacy, University of Santo Tomas, Manila 1015, Philippines; 5The Graduate School, University of Santo Tomas, Manila 1015, Philippines; 6Research Center for the Natural and Applied Sciences, University of Santo Tomas, Manila 1015, Philippines; 7Department of Aquatic Biosciences, National Chiayi University, 300 University Rd., Chiayi 60004, Taiwan; 8Department of Chemistry, Chinese Culture University, Taipei 11114, Taiwan; 9Research Center for Aquatic Toxicology and Pharmacology, Chung Yuan Christian University, Taoyuan 320314, Taiwan

**Keywords:** OpenCV, blood flow, zebrafish, medaka, Gunner Farneback algorithm

## Abstract

**Simple Summary:**

Artificial intelligence (AI) has been established to contribute in number of research areas such as medical imaging, diagnostic tools, ultrasound, cardiac scans, X-rays and blood flow analysis. OpenCV is a library having programming functions that provide ample of modules for digital image processing and artificial intelligence. In this study, we introduced a software package as OpenBloodFlow that can measure blood flow velocity and blood cell count precisely by selecting the dorsal aorta of zebrafish. The program is based on python programing language, which is a high-level, general-purpose language to solve biological problems. We present a cost-effective and fully automatic tool to perform the analysis as compared to commercially available software(s) that are not freely available to access. The key features of the program include a user-friendly graphical user interface (GUI), blood flow velocity and blood cell count in the given video dataset. The results are automatically saved in a CSV file without using any external plugins or third-party software and the program does not require GPU average CPU can be used for the execution.

**Abstract:**

The transparent appearance of fish embryos provides an excellent assessment feature for observing cardiovascular function in vivo. Previously, methods to conduct vascular function assessment were based on measuring blood-flow velocity using third-party software. In this study, we reported a simple software, free of costs and skills, called OpenBloodFlow, which can measure blood flow velocity and count blood cells in fish embryos for the first time. First, videos captured by high-speed CCD were processed for better image stabilization and contrast. Next, the optical flow of moving objects was extracted from the non-moving background in a frame-by-frame manner. Finally, blood flow velocity was calculated by the Gunner Farneback algorithm in Python. Data validation with zebrafish and medaka embryos in OpenBloodFlow was consistent with our previously published ImageJ-based method. We demonstrated consistent blood flow alterations by either OpenBloodFlow or ImageJ in the dorsal aorta of zebrafish embryos when exposed to either phenylhydrazine or ractopamine. In addition, we validated that OpenBloodFlow was able to conduct precise blood cell counting. In this study, we provide an easy and fully automatic programming for blood flow velocity calculation and blood cell counting that is useful for toxicology and pharmacology studies in fish.

## 1. Introduction

Blood flow velocity is one of the important parameters to assess the health of blood circulation, which is affected by many factors including blood vessel size, the viscosity of blood and heart pumping is controlled by the central nervous system [1]. Although this parameter is important, few methods have been established [2] for measuring non-invasive and reliable blood flow velocity in the low vertebrate model of zebrafish. Previously, most studies used commercial software of either MicroZebraLab™ [3] or DanioScope™ [4] to calculate the blood flow velocity in zebrafish with a robust calculation. However, since they were paid software, it is not always feasible to use for every laboratory due to limited resources. Therefore, our lab has established an ImageJ-based method to estimate blood flow velocity in zebrafish from 2D videos using a simple light microscope coupled with a high-speed camera [5]. Other previously published methods for blood flow measurement in zebrafish are also summarized in Table 1 for comparison. Nevertheless, the intensive labor and long steps needed to analyze the videos and more research to develop a fast, low-cost, fully automatic, and reliable non-invasive method for measuring blood flow velocity systems in fish.

The assessment of fish cardiac endpoints [12] and cardiovascular physiology [13] is complex and context dependent. In the world of computer vision, OpenCV has been contributing for more than 30 years [14]. The intense work related to optical flow has been done in gradient-based algorithms as a Spatio-temporal intensity module used to detect surface objects [15,16,17]. OpenCV provides numerous applications in the biomedical signal and medical image processing areas. The dimensions of analysis using OpenCV include optical systems and clinical diagnosis through X-ray, ultrasound, magnetic resonance imaging (MRI), nuclear medicine, and endoscopy [18]. Clinical examinations mostly rely on these well-known techniques to follow up the physiological changes in treatment and with help of AI number of advanced tools are available to investigate the clinical diagnosis. Python programming language has an efficient level-data structure to incorporate object-oriented programming (OOP) [19] and is compatible with working on modern operating systems such as Windows and Linux. OpenCV library contains more than 300 functions and is widely applicable in videos to track, segment, recognize and detect the moving objects [20]. OpenCV designs enable the user to implement real-time applications to capture object features and this advantage is versatile to use in medical applications. The bounding box theory in optical flow measurement has been well recognized for establishing the concept of the region of interest (ROI) for detecting moving objects [21]. It is challenging to determine the exact number of moving objects using traditional ROI-based algorithms [22]. Blood flow velocity measurement has computation barriers to detect the moving blood cells due to poor visualization in the real-time analysis [23]. In this study, we proposed a method by using an optical flow algorithm getting ROI selection that measures the blood flow velocity and blood cell count in a given video dataset.

Zebrafish is a suitable animal model for practicing because its translucent body enables researchers to perform studies as cardiac endpoints and blood velocity measurements. Fish have different blood flow velocities due to the size and structure of blood vessels in their circulatory system. Optical flow algorithms are designed to inquire about the displacement of moving objects using sensory and visualization mechanisms depending upon the acceleration of moving objects [24]. These algorithms are more precisely applicable to the video frames that provide mean displacement of the moving objects from one frame to another. Regarding powerful genetic and drug toxicity models, zebrafish and medaka are relatively convenient to use for cardiovascular studies [25]. The translucent embryo presents complete visibility to observe stages of embryogenesis. Zebrafish heart also contains comparable morphological features to mammals and humans while having close vascular anatomy to other vertebrates [26,27]. Moreover, the cardiovascular system in zebrafish can be observed with great clarity in a non-invasive manner, which makes data acquisition fast [28]. Similarly, medaka offers several advantages as a model organism for human disease and drug discovery studies similar to zebrafish [29].

The blood velocity analysis method using animal models is highly significant in biomedical research, and an automated approach for measuring blood flow velocity and the blood cell count is very useful. Previously, the ImageJ-based method was considered tedious and time-consuming for tracking blood cells. To overcome this disadvantage, this study aimed to automatically measure blood flow velocity with blood cell counts in selected ROI using cutting-edge computer technology. Additionally, several validation experiments were conducted to compare blood flow velocity measurement side-by-side using ImageJ- and OpenBloodFlow-based methods. Moreover, in our method, several videos can be batch analyzed which is considered a solid breakthrough to facilitate data processing speed to conduct cardiovascular physiology assessment.

## 2. Materials and Methods

### 2.1. Zebrafish and Medaka Maintenance and Embryo Collection

This study used wild-type AB strain zebrafish (*Danio rerio*) and wild-type Japanese medaka (*Oryzias latipes*) as experimental animals. Both fish were maintained in a continuously aerated water system. The temperature was maintained at 26 °C with 10/14 h of dark/light cycle. The zebrafish and medaka maintenance was performed according to the previous protocols [30,31]. All experiments were performed following the approval by the International Animal Care and Use Committees (IACUCs) of Chung Yuan Christian University (Approval No. 109001, issue date 15 January 2020).

To collect the embryos, sexually matured zebrafish with a 2:1 male/female ratio were put into the breeding chamber at night. The following day, the separator was removed, and the embryos were collected two hours after. The collected embryos were washed with distilled water and kept in an incubator at 28 °C until further experimentation. On the other hand, the medaka embryos were collected every morning and were labeled as 0 h post-fertilization (hpf) for synchronization of the time. The embryos were washed using distilled water and were kept in an incubator at 28 °C until further experimentation.

### 2.2. Chemical Treatment

Phenylhydrazine (PHZ) and ractopamine (RAC) (Shanghai Macklin Biochemical Co., Ltd., Shanghai, China) were diluted as a stock solution of 1000 ppm (*w*/*v*) by using 100% DMSO as a solvent and were further diluted using ddH_2_O. Zebrafish larvae at 48 hpf were treated with either 0.15 ppm (*w*/*v*) PHZ or 4 ppm RAC for 24 h and the corresponding blood flow alterations measured at 72 hpf. The control group was treated with 0.05% of DMSO as the solvent control and the final DMSO concentration in the treatment group was lower than 0.5%, which is still acceptable, according to the previous study [32,33].

### 2.3. Zebrafish and Medaka Video Processing

For blood vessel video acquisition, zebrafish embryos aged 2-, 3-, 4-, and 5-days post fertilization (dpf) and medaka embryos aged 10 dpf were used and mounted with 3% methylcellulose solution to immobilize body movement. A high-speed digital charged coupled device (CCD) camera (AZ Instrument, Taichung City, Taiwan), mounted on an inverted microscope (ICX41, Sunny Optical Technology, Yuyao, China) was used to record high-speed videos. In addition, a Hoffman modulation objective lens at 40× was used to enhance image contrast for blood cells. The recording was focused on the dorsal aorta in the trunk area, and the recording was done by HiBestViewer software (AZ Instrument, Taichung, Taiwan) to obtain a 10-s short video with a frame rate of 200 frames per second (fps).

### 2.4. Video Stabilization

The overall schema of the experimental design is summarized in Figure 1. The first step was to stabilize the video dataset of fish embryos, which is especially important for keeping fish embryos in a fixed position to reduce shaky movements. The input video format was mp4 and analyzed using the OpenCV module vidstab [34]. The stabilization module in OpenCV was used to capture the interframe information. Later, optical flow metrics, such as rigid Euclidean transformation, were used to show real-time video stabilization in shaky frames [35]. The new transformed video in a separate window is exhibited (Video S1) by using a sliding window smoothing trajectory. The moving objects as blood cells were masked using the OpenCV function to block the rest of the area other than the dorsal aorta [36]. A separate window was initiated for the masking in addition to the stabilization window for a better view.

### 2.5. Python Processing Modules

While Python is common for using parallel operations to achieve the desired task for a lot of crunching modules, such as pyautogui (https://pyautogui.readthedocs.io/en/latest/ (accessed on 21 December 2021)), math (https://docs.python.org/3/library/math.html (accessed on 21 December 2021)), matplotlib (https://matplotlib.org/ (accessed on 21 December 2021)), threadPool (https://docs.microsoft.com/en-us/dotnet/api/system.threading.threadpool?view=net-5.0 (accessed on 21 December 2021), and deque (https://www.geeksforgeeks.org/deque-in-python/ (accessed on 21 December 2021)) were applied in this study to measure blood flow in zebrafish and medaka. Nested parallelism was exposed on all possible levels to recognize the structural incidences in loops and functions to pipeline the frame motion in the video dataset [37]. The Pyautogui module was used to control the keyboard and mouse for the processing of videos. Deque module was used for multi-threading to apply multi-processing [38,39], and the computational power efficiency was utilized to speed up the results. These modules provided enriching blood flow analysis and avoided the creation of subclasses to process the calling function for respective operation directly. The Gunner Farneback optical flow algorithm was applied to measure the displacement in adjacent video frames [40,41]. Separate measurement scales were introduced for two types of fish data. To avoid exhaustive computational labor, ThreadPool [42,43] class was initiated.

While medaka and zebrafish have different blood velocities, to recognize the best level, SciPy (https://scipy.org/ (accessed on 23 December 2021)) library was used, and nested parallelism (https://github.com/esa/pygmo2/issues/24 (accessed on 23 December 2021)) was applied for combining blood velocity and blood cell count results in more than one thread. The basic concept of applying these two modules is to call functions from a parallel region to another region inside the script. The regular programming practice with Scipy [44] packages was used to deal with the numerical problems.

### 2.6. Blood flow Velocity Measurement Algorithm

In measuring the blood flow velocity in the video dataset of fish embryos, we used a previous measurement scale defined by our team based on the ImageJ platform [45]. The current approach differs from the previous ImageJ-based, one because it can automatically measure blood flow velocity and approximate number of moving blood cells compared to ImageJ. The unit scale defined in this study was millimeters per second, followed by the principle of optical flow measurement. First, we selected a video to initialize the process and select ROI at the dorsal aorta in the input video. Then, we applied the Opencv.createTrackbar (https://docs.opencv.org/3.4/da/d6a/tutorial_trackbar.html (accessed on 6 May 2022)) function, which is helpful to tweak a variable value instantly without closing and relaunching the program. By using this approach, we can read the current position of the trackbar slider that works in OpenCV [46]. Next, the selected ROI on each frame was processed to get blood flow velocity and average blood cell count in the video dataset. For better performance, we used calcOpticalFlowFarneback [47], np.linalg.norm [48], cv2.normalize [49] and cv2.threshold [50] functions.

CalcOpticalFlowFarneback [51] computes a dense optical flow under the principle of Gunnar Farneback’s algorithm. Due to the viscosity property in the fish blood, the Gunner Farneback algorithm was used to compute optical flow [52] for all the points in the frame instead of sparse feature computation in the Lucas-Kanade algorithm [53]. As per frame, optical flow attempts to determine where each pixel is being shifted. The practiced grayscale images are mainly used as input, followed by the traditional logic that applied in video frames analysis. Traditionally, programming practice was in larger values, and the increased robustness of the algorithm showed image noise and yielded blur motion field. To overcome this problem, np.linalg.norm [54] function was applied. To set the order to None as default and to calculate the Frobenius norm [55], we implemented normalization of streaming of blood flow velocity in consecutive frames. By using cv2 normalization [56], the changing pixel intensity formula applied to increase the overall contrast. We used cv2.threshold function to convert the color image to the binary image, which is an efficient and well-known technique in image processing. In addition, we applied average optical flow and counted analytic thresholds in moving objects. Finally, oscillation graphs [57] were obtained to show overall blood flow velocity and blood cell count in all video frames. Graph smoothing was performed using the SciPy library in python.

### 2.7. Computation of Blood Flow Velocity

The computation of blood flow velocity in 2D videos was performed using relative frame estimation metrics in the motion of a moving object. As the pixel moves to a certain distance in the video, the time series based on fps is satisfied following the equation [58]:(1)$$I\left(x,y,t\right)=I\left(x+dx,y+dy,\;t+dt\right)$$
where I denoted as pixel intensity, x,y are the vectors to calculate the average velocity of blood at fps as *t* and movement of the pixel at a distance $\left(dx,dy\right)$. Each time increment was shown as *dt*, and approximation was calculated using the Taylor series [59]:(2)$$f_{x}u+f_{y}v+f_{t}=0$$
(3)$$f_{x}=\frac{\partial f}{\partial x};\;f_{y}=\frac{\partial f}{\partial y}$$
(4)$$u=\frac{dx}{dt};v=\frac{dy}{dt}$$
where $f_{x}$ and $f_{y}$ are the gradients of the frame, and ft  is the gradient along time with *u* and *v* vectors.

There are other methods to compute optical flow, such as Horn-Schunck, Buxton [59], Black-Jepson [60], Lucas-Kanade and Farne and Farneback. In this study, the Gunner Farneback algorithm was implemented since it was well-known for testing the two frames estimation in polynomial expansion. The blood flow velocity was categorized into an average ratio to determine the maximum displacement in frames. Two separate scales were defined in the program to execute two different videos dataset by adjusting the range for normal zebrafish embryos from 0.3 to 1.5 mm/s, followed by the general values recorded for 2–5 dpf. This is because OpenCV can analyze the velocity of moving objects at a time (second) in selected ROI. For medaka, the scale was defined by computing the blood flow velocity in ImageJ, and settings were adjusted by creating two input values as an option to select 1 for medaka and 0 for the zebrafish to get results for two different types of fish. This feature does not require in case of using the program with GUI.

### 2.8. Implementation of Algorithm

The consecutive frames difference was computed in the selected ROI of zebrafish and medaka videos. Contours were detected by using OpenCV in all joining points of the frame which have the same intensity [61]. The motion field estimation in 2D videos was combined with horizontal and vertical optical flow components. To get the blood flow velocity in uniform analysis, the direction of contours was defined as per Gurav and Kadbe’s method [62]. Gunnar Farneback algorithm is highly extensive in estimating dense optical flow and the computation of this method is slower than the Lucas-Kanade method; however, its accuracy and results are more conducive. The algorithm detects a relative change in pixel intensity between the two images using polynomial expansions and shows the pixels with the most significant change. The method depends on the approximation of neighborhood pixels in relative frames. The displacement in frames was used to count in the transformation of polynomials.

As per equations 1 to 4, results can be satisfied by equating an iterative pixel solution. This study applied the principle with corresponding high values in frame approximation. The quadratic polynomial expression is defined with adjacent frames at times *t*_1_ and *t*_2_ as follows [63]:(5)$$f_{1}\;\left(x\right)=x^{T}A_{1}x+b_{1}^{T}x+c_{1}$$

*A* presenting symmetric matrix, *b* is the vector, and *c* is used as a scalar. To consider respective quadratic polynomials, a refined signal f2 was  constructed for the global displacement (*d*) [64]. Using the global displacement concept shown in Equations (6) and (7), we equated the pixel with an iterative result to get the average blood flow velocity in each zebrafish video.
(6)$$\begin{array}{cc} f_{2}\left(x\right)=f_{1}\left(x-d\right) & ={\left(x-d\right)}^{T}A_{1}\left(x-d\right)+b_{1}^{T}\left(x-d\right)+c\\ & =x^{T}A_{1}x+{\left(b_{1}-2A_{1}d\right)}^{T}x+d^{T}A_{1}d-b_{1}^{T}d+c_{1}=x^{T}A_{2}x+b_{2}^{T}x+c_{2} \end{array}$$
(7)d=12A1−1b2−b1

### 2.9. Blood Cell Counting

The blood cell count is an important feature of this study. The method was tested using two directional optical flow standards to measure the blood cells concerning the contours in the selected ROI. In this study, the blood cells of medaka and zebrafish were computed based on velocity per frame and the number of blood cells positioned next to each other.

Later, the calculation was enlarged to the width of ROI to get an average of how many rows of blood cells were present. Then, the total number of blood cells was counted for each second. Get velocity/frame = velocity/frames_interval_value, as we have the velocity in frames that we see in millimeters, instead of seconds. Then we assigned *count*_1_*sec* = 0 as we take the computation until the time is 1 s.
(8)$$\mathrm{count}\_1\_\sec=\frac{\mathrm{count}\;1\;\sec+\left(\mathrm{avg}\_\mathrm{blood}\_\mathrm{cells}\_\mathrm{rows}\times \mathrm{avg}\_\mathrm{cell}\_\mathrm{width}\right)}{\mathrm{velocity}\_\mathrm{per}\_\mathrm{frame}}$$

To determine the relative blood cell count for one second, we used the average from the total number of blood cells passing from each frame. The limitation of OpenCV is that it cannot detect 100% moving blood cells in ROI for each frame. We used an automated feature to get ROI using the OpenCV cv.drawContours module [65]. The average blood cell count was performed by extracting the per-frame data in an array and then dividing the sum by the time of video processing (1 min) approximately. The total blood cell count was dependent on the time and the velocity of moving blood cells in each frame. For data validation, we manually counted the blood cells passing through the video frames in a horizontal direction (left to right) using the naked eye. We validated a total of 20 fish embryos aged at different development stages. For manual counting, 10 frames were selected at a fixed interval for a specific fish embryo and we validated 20 fish embryos for each developmental stage. By using the following equations, the horizontal direction flow of blood was computed:(9)$$H\;\left(x,y\right)=h_{1}\left(x,y\right)+h_{2}\left(x,y\right)$$
(10)$$V\left(x,y\right)=v_{1}\left(x,y\right)+v_{2}\left(x,y\right)$$

*H* and *V* are the frames of horizontal and vertical optical flow while (x,y) are the pixel coordinates. The pixel value was set as per grey scale (0, 255) for 8-bit greyscale images and each cycle of blood streaming in the video was observed in automated ROI. The success of unidirectional blood cell counts under optical flow was obtained. To get the stable oscillation for blood cell count, we used distinction moving contours that are being calculated in binary images using the traditional data structure approach for contour listing. The contours defining the blood cell counting method were implemented similarly to Suzuki [66] and Abe’s approach [67]. It has been observed during analysis that automatic ROI failed to detect the blood cell count for some videos with poor resolution, and cells were covered with masking. In those videos with low quality or poor contrast, the blood cell count was underestimated by using OpenBloodFlow. To solve this problem, we developed a multitasking Python feature to get manual ROI for the blood cell count in the dorsal aorta. Two options would be displayed in the OpenBloodFlow platform for users to select either the automatic ROI feature or the manual ROI option. Both scenarios would work efficiently, and manual ROI selection could help to overcome the problem of capturing missing blood cells in a given video dataset.

### 2.10. Statistical Test

Statistical analysis was done using GraphPad Prism (GraphPad Inc., La Jolla, CA, USA). Normality and relative standard deviation measurements were performed to select the appropriate statistical test. Either parametric or non-parametric tests were performed according to the normality of data distribution. The variance of the data and the significance were calculated based on the appropriate post-hoc multiple comparison test.

### 2.11. Graphical User Interface Designing

The GUI was also developed in this study to achieve a more user-friendly platform than the command line process to save and follow up the results. The original GUI was structured in Python, and the basic idea involved uploading the folder containing videos and starting the analysis with a single click. For GUI development, we used tkinter (https://docs.python.org/3/library/tkinter.html (accessed on 3 June 2022)) python library. This tkinter library is widely used to create GUI-based desktop applications for Windows and Linux operating systems. We applied the Canvas for video control with tkinter.Tk function and masked window having dorsal aorta view, placed in the center of the main GUI window. Four functional buttons were created as presented in Figure 2. For selecting the video folder filedialog.askdirectory function was used. The selection of ROI was designed with two choices, automatic and manual options presenting to run function and execute the program. The final frame was processed to generate the image of the dorsal aorta in the video, aligned in the center of the GUI window. Finally, the result button was designed to show the folder having oscillations of each video’s blood flow velocity and blood cell count on average.

## 3. Results

### 3.1. Overview of Analysis Pipeline for Blood Flow Measurement

In a previous study, our team developed an ImageJ-based method to manually compute blood velocity and blood cell tracking in zebrafish embryos. To reduce the operational complexity of the previous method, we aimed to develop an efficient and automatic method for measuring blood flow velocity and blood cell count for zebrafish and medaka fish embryos.

In this study, the new platform is based on OpenCV, which provides many convenient modules for video analysis (analysis pipeline summarized in Figure 1A) presented. Due to fast blood flow velocity, a camera with the capability to record at a high frame rate (200 frames per second, fps) was preferred to record the dorsal aorta region to minimize the loss of information. The original video was transformed to 30 fps with slow motion having a 6.6-fold slowdown effect. The recorded videos were processed with the OpenCV program in Python. In this study, the dorsal aorta region was selected as the targeted ROI. The rectangular feature of ROI selection integrated with a separated window shows the dorsal aorta in a masked region under video stabilization. Figure 1B,C show the difference between the original and stabilized videos. The video stabilization aims to increase the quality and remove the shaking movements of the fish in the video. The local motion estimation focused on the dorsal aorta view, and masking was performed to better visualize a stabilized result compared to the original (destabilized) video. Figure 1D,E shows the motion illustration in estimating optical flow using the Gunnar Farneback method in OpenCV. After the methodology was established, we conducted validation by making a side-by-side comparison between OpenCV and ImageJ (or manual counting) to calculate zebrafish and medaka blood flow velocity and blood cell count respectively.

### 3.2. Easy Operation of OpenCV to Measure Zebrafish Blood Flow

To reduce the operational difficulty, we developed a user-friendly GUI package OpenBloodFlow, to conduct blood flow velocity and blood cell count measurements in the dorsal aorta of zebrafish and medaka embryos (Figure 2). We used zebrafish embryos as a model to demonstrate the operation and the data outlook. After video uploading and ROI selection in the dorsal aorta region (Figure 3A), the OpenBloodFlow can automatically mask the area with barely any motion change from frame-to-frame manner (Figure 3B). This masking step is important to extract dynamic blood flow information precisely. By optical flow analysis with the Gunnar Farneback method, the OpenBloodFlow program could generate good quality figures reporting the average blood flow velocity (Figure 3C, left panel) and the average blood cell count (Figure 3C, right panel). All this process can be finished within 120 s since 66 s-long videos had been converted first from 200 to 30 fps. In the current program setting and video recording, we found OpenBloodFlow auto ROI function can precisely select the entire dorsal aorta with 85% accuracy (68/80 videos tested). The other 15% of videos with low quality need to conduct manual ROI selection to avoid potential underestimation problems for blood cell count. The OpenBloodFlow GUI package and user manual were provided as Appendix A.

For performance comparison in blood flow velocity calculation, we analyzed videos using either ImageJ or OpenBloodFlow in the same system environment of Intel Core™ i9-9900KF CPU processor with 32.0 GB RAM. Results showed better performance for OpenBloodFlow than ImageJ due to its relatively faster and consistent calculation speed on the contrary, high inter-video variations and relatively two times longer analysis time were detected in the ImageJ method (Appendix B Figure A1).

### 3.3. Methodology Validation Case 1: Blood Flow Measurement in Zebrafish Larvae at Different Ontological Stages

To validate the utility of the OpenBloodFlow GUI package, we initially compared the blood flow velocity in zebrafish embryos at different developmental stages using either OpenCV or ImageJ methods. Video datasets for zebrafish aged at 2, 3, 4, and 5 dpf (day post-fertilization) were obtained from our previous publication. Later, the same video dataset was analyzed by both OpenCV and ImageJ in parallel, and the statistical significance was determined by a paired *t*-test. In 2 dpf, the average blood flow velocity was 386.9 ± 144.4 µm/s for OpenCV and 394 ± 133.9 µm/s for ImageJ methods (Figure 4A). In 3 dpf, the average blood flow velocity was measured as 418.2 ± 123.4 µm/s for OpenCV and 375.3 ± 90.04 µm/s for ImageJ methods (Figure 4B). In 4 dpf, the average blood flow velocity was calculated as 480.3 ± 153.9 µm/s for OpenCV and 542 ± 131.7 µm/s for ImageJ methods (Figure 4C). In 5 dpf, the average blood flow velocity was measured as 505.4 ± 151.6 µm/s for OpenCV and 569.9 ± 134 µm/s for ImageJ methods (Figure 4D). Despite gaps in values obtained per age group, paired *t*-test results showed no significant difference in every age group between OpenCV and ImageJ methods (2 dpf, *p* = 0.6743; 3 dpf, *p* = 0.0712; 4 dpf, *p* = 0.0779, and 5 dpf, *p* = 0.0551). Those results show that the average blood flow velocity obtained by OpenCV is consistent with ImageJ from 2 to 5 dpf.

### 3.4. Methodology Validation Case 2: Comparison of Blood Flow Velocity in Zebrafish after PHZ Exposure

To validate the OpenCV method for detecting blood flow alteration, we induced thrombosis in zebrafish embryos using PHZ (phenylhydrazine) to reduce blood flow velocity. PHZ is a chemical compound that can cause thrombosis in several animal models like rats and fish. In line with a previous study [68], a significant decrease in blood flow was observed after incubation with PHZ at 0.15 ppm (*p* < 0.0001, Figure 5C). Compared to control embryos aged at 3 dpf, the blood flow in PHZ-exposed embryos sharply decreased from 349.3 ± 80.3 to 177.9 ± 41.5 µm/s as detected by OpenCV, and from 359.6 ± 89.4 to 175.4 ± 39.2 µm/s via ImageJ methods. Paired *t*-test displays no significant difference for OpenCV and ImageJ methods for the control group (*p* = 0.5201) or PHZ-exposed group (*p* = 0.1827) (Figure 5C). Those results show that OpenCV can be used to detect reduced or slow blood flow velocity induced by PHZ, consistent with the results obtained from the ImageJ calculation.

### 3.5. Methodology Validation Case 3: Comparison of Blood Flow Velocity in Zebrafish after RAC Exposure

To validate the OpenCV method for blood flow velocity detection, we used RAC (ractopamine), a beta-adrenoreceptors agonist, to elevate oxygen consumption, locomotor activities, and blood flow velocity in zebrafish. In line with a previous study [69], a significant increase in blood flow was observed following incubation with RAC at 4 ppm by either ImageJ (*p* < 0.001, Figure 6A) or OpenCV (*p* < 0.001, Figure 6B) method. Validation by paired *t*-test display no differences between OpenCV and ImageJ methods for either control (*p* = 0.424) or RAC-exposed group (*p* = 0.616) (Figure 6C). Those results show OpenCV method can be used to analyze elevated blood flow velocity induced by RAC, comparably consistent with the results of the ImageJ method.

### 3.6. Methodology Validation Case 4: Comparison of Blood Flow Velocity in Medaka

Next, we checked the versatility of the OpenCV method in Japanese Medaka fish (*Oryzias latipes),* another important freshwater fish model for ecotoxicity studies. The newly hatched Japanese medaka fish embryos aged 10 dpf were subjected to blood flow velocity measurement using the OpenCV or ImageJ methods (Appendix B Figure A2A). The target ROI was selected within the area of the dorsal aorta, similar to the position used in zebrafish (Appendix B Figure A2B). The OpenCV measured a blood flow velocity of 402.3 ± 87.16 µm/s and 419.9 ± 79.06 µm/s for the ImageJ method (Appendix B Figure A2C). These values are not significantly different according to paired *t*-test (*p* = 0.136). This result demonstrated the versatility of our advanced OpenCV method in analyzing the blood flow velocity of other fish like medaka.

### 3.7. Methodology Validation Case 5: Blood Cell Count Validation

For the blood cell count function in the OpenBloodFlow GUI package, we conducted performance validation by comparing it with the data obtained from manual counting. For manual counting, we measured the average blood cell count in 20 zebrafish embryos (each embryo with 10 frames) aged at 2, 3, 4, and 5 dpf and compared the average values to OpenBloodFlow results. Results showed that the blood cell count obtained from OpenBloodFlow is not significantly different from the results of manual counting in 2 dpf (65 ± 25 vs. 69 ± 26 cell/frame, *p* = 0.9931), 3 dpf (76 ± 13 vs. 79 ± 13 cells/frame, *p* = 0.9991), 4 dpf (44 ± 10 vs. 50 ± 11 cells/frame, *p* = 0.9832) and 5 dpf (33 ± 16 vs. 37 ± 17 cells/frame, *p* = 0.9975) embryos after one-way ANOVA statistical analysis (Figure 7). In addition, the average blood cell count per frame increased at 2 to 3 dpf and gradually declined at 4 and 5 dpf (Figure 7). At the time of writing, no detailed prior studies in the field have achieved this. However, this might have happened because of the differentiation that happen at that time that might change the composition of blood cells, but more studies have to be done to support this hypothesis. Overall, the OpenBloodFlow established in this study indeed can be used to measure blood cell count in zebrafish embryos in a relatively precise manner as supported by manual counting.

## 4. Discussion

Digital image processing approaches have been widely applied in examining fish physiology and development in a non-invasive manner. The most important finding in this study is that we offer a simple, cost-effective, and fully automated tool for blood flow velocity and blood cell counting in zebrafish and medaka by employing an OpenCV-based approach for the first time. With detailed and careful validation, this OpenCV-based tool called OpenBloodFlow indeed can be used for blood flow velocity and blood cell count measurement in both important fish species that can be widely applied for toxicology, pharmacology, and drug screening studies. Batch processing is an additional feature of this tool to analyze multiple videos at once, which may reduce the effort of selecting data one by one manually. The following section presents the potential Pros and Cons of our newly established OpenBloodFlow GUI package [70].

### 4.1. Advantage of Current Reported OpenCV Method

NIH initially designed ImageJ to conduct image editing and measurement as an open-access platform. Although ImageJ has expanded its function, it is less powerful than OpenCV in video analysis [71]. Compared to ImageJ, OpenCV consists of more functions and modules that are well established in image processing to solve critical image processing problems [72,73]. It provides many visual processing features that can assist users in compiling heavy datasets in an executive programming environment in Python. It has been widely applied in Human-Computer Interaction (HCI) object recognition, image segmentation, motion tracking, and object tracking [74].

Compared to previous techniques, our study provides an innovative and fully automatic approach to detecting blood flow velocity and blood cell count together in a real-time video by using the OpenCV approach. The most effective application of our OpenBloodFlow tool can precisely detect the blood flow alterations after chemical treatment which provides a very important tool for toxicology and pharmacology studies by using fish embryos as a simple model. Finally, by calculating the geometric area mean [75] using the GetOptimalDFTSize function, we can deduce the average count of blood cells in this OpenBloodFlow GUI package. To make the program efficient and process multiple videos to save analysis time, there was a loop-wise operation [76] was performed for the number of videos inputted at once. In addition, the data calculation is also about 2-fold faster than our previously reported ImageJ-based method (Appendix B Figure A1).

Key features involved in this study do not require fluorescence probes/fish or heavy mechanical microscopic equipment to measure blood flow velocity and blood cell count in zebrafish or medaka which makes it applicable for low- or middle-sized laboratories. Furthermore, ordinary CCD videography in the current study exhibited good results compared to other methods that used expensive equipment. Schwerte et al. [77] have previously reported blood cell count in zebrafish based on greyscale value and motion in the frame. Compared to Schwerte’s method, our method does not necessitate numerical transformation for greyscale values, and it also provides a more user-friendly interface for user operation in a fully automated manner. A summary of the available software package to measure blood flow velocity is available in Table 2.

### 4.2. Potential Limitations and Future Work

There are some potential limitations in the current version of OpenBloodFlow that require more studies to address these limitations. First, video recording quality plays a significant role in measuring blood flow velocity and blood cell count. The poor resolution would reflect incorrect ROI selection at blood cells. However, this problem does not affect the blood velocity measurement. If the video has poor quality contrast to differentiate moving blood cells, the automated ROI selection function cannot select the whole blood vessel as dorsal aorta. It might get partial ROI selected in the trunk area, which reduces the blood cell count value but not the blood velocity. Second, another scenario that may potentially reduce the blood cell count performance of OpenCV is cell overlapping. This problem is challenging to solve based on the OpenCV method. Future studies that will try to develop better cell segmentation and objective recognition features might be able to overcome this limitation. Third, the current cell counting method for fish blood vessels is based on 2D videography. To get more reliable data, 3D image construction by advanced microscopic techniques for real-time blood flow velocity with an exact number of moving blood cells is considered necessarily. Finally, in the current version, two different scale settings are incorporated based on the intensity of moving blood cells in videos of zebrafish and medaka. The sensitivity of the recent study is based on slight movements of blood cells that are restricted in the ROI boundary line for all frames. More efforts are required in the future to adjust the algorithm setting for a more unified scale setting that can measure blood flow at different speeds.

## 5. Conclusions

The concept used in this study was to analyze blood flow velocity and blood cell count in zebrafish and medaka embryos using versatile and user-friendly OpenCV-based software. Video stabilization was used to normalize the shaky video footage of the dorsal aorta. The positioning of blood cells in the video was attempted to describe the tracking in the desired contour. The relative positions were determined based on two adjacent frame differences. The study implemented the Gunner Farneback algorithm to determine optical flow and compute the blood flow velocity in the zebrafish and medaka videos dataset. The utility of the Gunner Farneback algorithm is consistent in blood flow velocity measurement and the number of levels at each labeling of scales recorded in millimeters per second unit. The oscillation of blood flow velocity and blood cell count are shown in separate graphs. Overall performance of the tool was tested by comparing the operation time side by side with the ImageJ software in the same environment system. The analysis showed that OpenBloodFlow has faster processing compared to the ImageJ method. This study proposed a convenient method to compute blood flow velocity and blood cell count in zebrafish and medaka embryos to ease the burden of the manual job and semi-automated processing. In the future, it is suggested that this approach might assist professionals, researchers, and medical practitioners in analyzing thrombosis and blood flow velocity problems for clinical findings.

## Figures and Tables

**Figure 1 biology-11-01471-f001:**
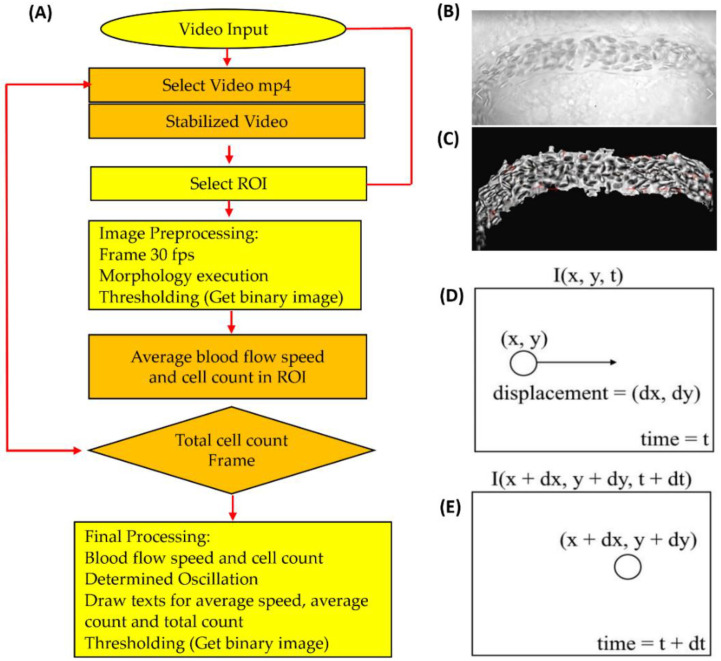
The overall scheme for blood flow velocity and blood cell count in zebrafish and medaka. (**A**) Overview of the entire analysis pipeline for blood flow measurement in fish embryos by OpenCV. (**B**) Shaky video footage of a fish blood vessel. (**C**) After performing video stabilization, the masked area presented for smooth oscillation. The motion illustration in estimating optical flow was applied using the Gunnar Farneback method by OpenCV at time = t (**D**) and time = t + dt (**E**).

**Figure 2 biology-11-01471-f002:**
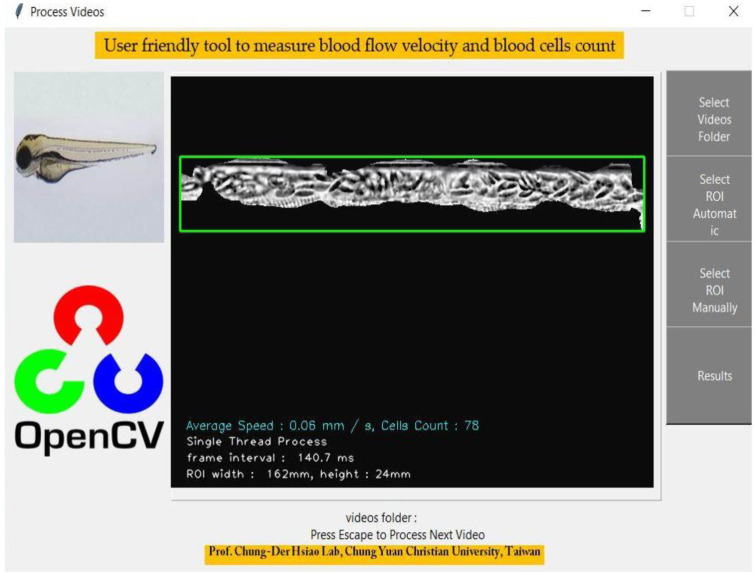
Simple graphical user interface of OpenBloodFlow GUI Package with complete functions for blood flow velocity and blood cell count analysis. OpenBloodFlow can automatically or manually select the blood vessel as dorsal aorta depending on video contrast and quality. We suggest using the manual function to determine specific ROI for blood cell count analysis for low contrast or poor videos quality.

**Figure 3 biology-11-01471-f003:**
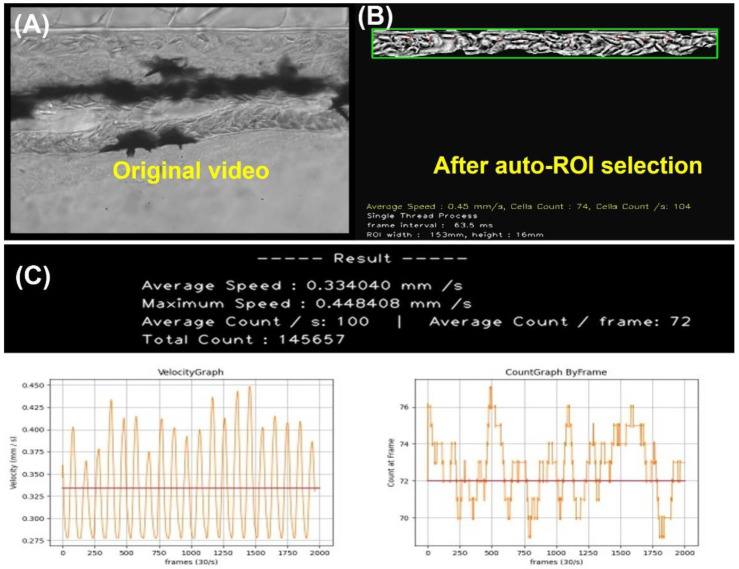
The computation of blood velocity and blood cell count in selected ROI and the displayed oscillation pattern over time in zebrafish aged 3 dpf. (**A**) ROI was chosen within the dorsal aorta for blood flow measurement. (**B**) Shaky video footage of blood vessels after performing video stabilization and the masked area presented for smooth oscillation. (**C**) The red line points to the average blood flow velocity (left panel) and blood cell count (right panel).

**Figure 4 biology-11-01471-f004:**
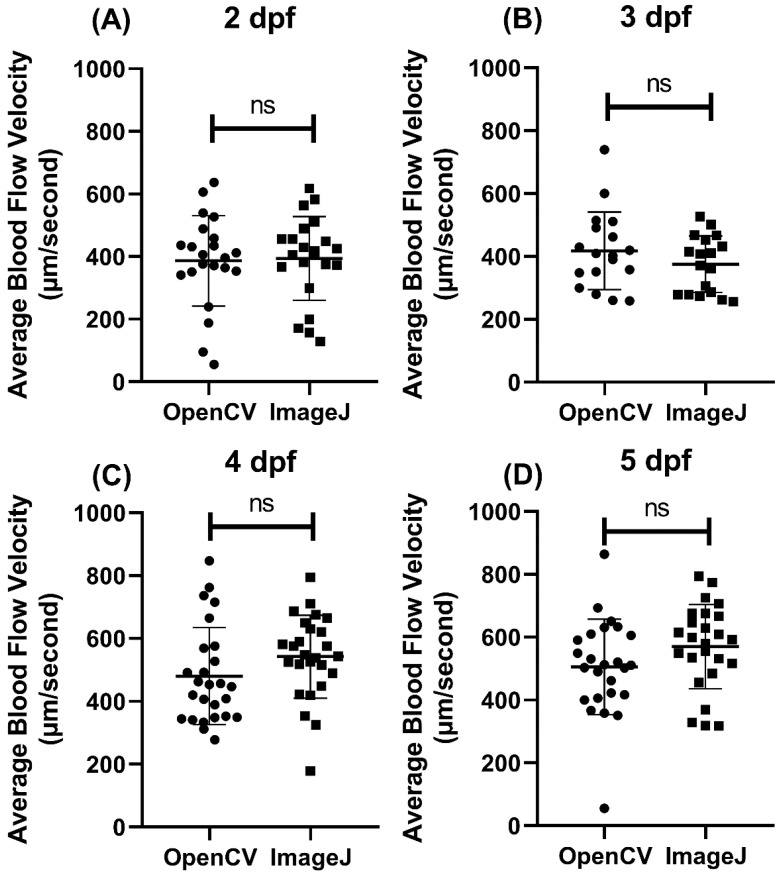
Validation of the performance of OpenBloodFlow GUI package for blood flow velocity measurement through comparison with ImageJ method. Zebrafish embryos aged (**A**) 2 dpf, (**B**) 3 dpf, (**C**) 4 dpf, or (**D**) 5 dpf were subjected to ImageJ and OpenCV methods to measure the average blood flow velocity. The data were shown as mean with standard deviation (circle dots show the data analyzed by OpenCV, while square dots show the data analyzed by ImageJ), and the significant difference was calculated by paired *t*-test. (ns = no significance, *n* = 18–26).

**Figure 5 biology-11-01471-f005:**
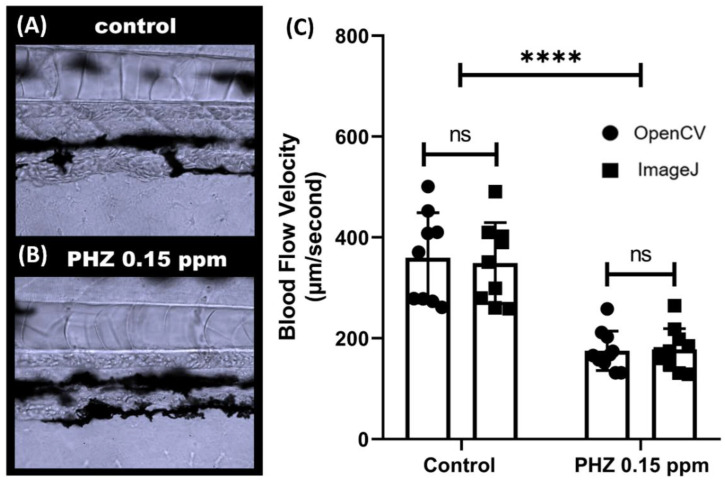
Validation of OpenBloodFlow method through comparison with ImageJ performance for blood flow velocity measurement in zebrafish larvae after exposure to control (**A**) and 0.15 ppm PHZ (Phenylhydrazine) (**B**). PHZ exposure significantly reduced the blood flow velocity in zebrafish larvae. (**C**) Side-by-side comparison of the average blood flow velocity measurement by ImageJ and OpenCV methods. The data were presented as mean with standard deviation, and statistical significance was determined by paired *t*-test for intra-group comparison or unpaired *t*-test for inter-group comparison. (ns = no significance, **** *p* < 0.0001, *n* = 10) (circle dots show the data analyzed by OpenCV, while square dots show the data analyzed by ImageJ).

**Figure 6 biology-11-01471-f006:**
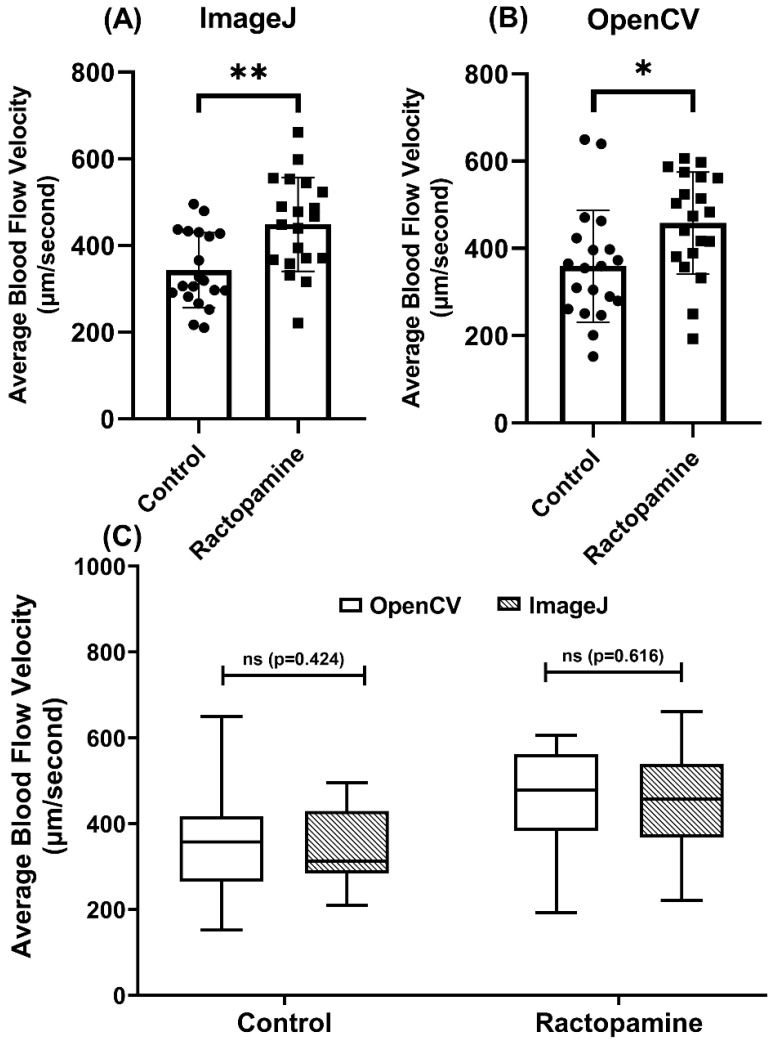
Validation of OpenBloodFlow performance for blood flow velocity measurement in zebrafish larvae through comparison with ImageJ method. Average blood flow velocity measured by ImageJ (**A**) and OpenBloodFlow (**B**) after exposure to ractopamine. Ractopamine exposure significantly elevated the average blood flow in zebrafish. (**C**) Side-by-side comparison of the average blood flow velocity measurement by ImageJ or OpenCV methods. The data were shown as mean with standard deviation and statistical significance determined by paired *t*-test for intra-group comparison (ns = no significance, * *p* < 0.05, ** *p* < 0.01, *n* = 20) (circle dots show the control data, while square dots show the ractopamine data in A and B).

**Figure 7 biology-11-01471-f007:**
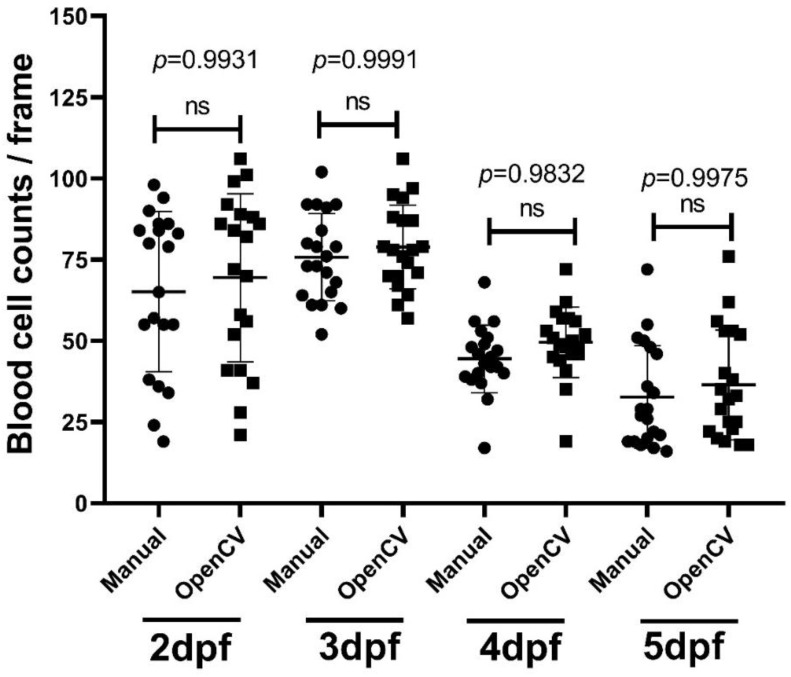
Validation of blood cell count in zebrafish embryos calculated by OpenBloodFlow through comparison with manual counting method. Zebrafish embryos aged at 2 dpf, 3 dpf, 4 dpf, or 5 dpf were subjected to blood cell counting using the naked eye (Manual) and OpenCV method. The data were shown as mean with standard deviation, and statistical significance was determined by an ordinary one-way ANOVA test (ns = no significance, *n* = 20). (circle dots show the data analyzed manually by manual counting, while square dots show the data analyzed by OpenCV).

**Table 1 biology-11-01471-t001:** Comparison of the previous methods used to detect blood flow velocity and associated endpoints in zebrafish.

Author and Publication Year	Major Facility to Capture Heartbeat Images	Measurement Principle	Region of Interests (ROI)	Endpoints Measured
Santoso et al. (2019) [5]	High-speed camera with an inverted microscope	Dynamic pixel changes over time	Dorsal Aorta, Posterior Cardinal Vein	Blood flow velocity, stroke volume
Yeo et al. (2019) [6]	Custom-built, 64-channel high-frequency array imaging system and a high-frequency linear array transducer with 256 elements	Pulsed wave spectral Doppler imaging	Heart, dorsal aorta	Blood flow velocity, Heart regeneration
Chiang et al. (2020) [7]	A 70-MHz ultrasound imaging system and single-element transducer	2D autocorrelation velocity estimation algorithm	Heart, dorsal aorta	Blood flow, tissue velocity, and cardiac deformation measurement
Parker et al. (2014) [8]	High-speed camera with an inverted microscope	Change in pixel density on cardiac muscles area	Dorsal Aorta, Posterior Cardinal Vein	Blood flow velocity, heart rate
Zickus and Taylor (2018) [9]	SPIM-μPIV (Selective plane illumination microscopy combined with Micro-particle image velocimetry)	Fluorescence imaging over interrogation windows to get a correlation	Dorsal Aorta, Posterior Cardinal Vein	Blood flow velocity, stroke volume
Watkins et al. (2012) [10,11]	Inverted Fluorescence Microscope with Hamamatsu Flash 2.8 CMOS Camera	Subarray pixel differences over time	Dorsal Aorta	Blood flow velocity
This study	High-speed digital charged coupled Device with an inverted microscope	Dense optical flow measurement algorithm	Dorsal Aorta	Blood flow velocity and blood cells count

**Table 2 biology-11-01471-t002:** Comparison of available software packages to measure blood flow velocity or blood cell count in zebrafish.

Software Name	ROI Selection	Availability	Batch Processing
MicroZebraLab	Manual	Paid software	No
Danioscope	Manual	Paid software	No
Trackmate ImageJ	Manual	Freeware	No
OpenBloodFlow (This study)	Automatic	Freeware	Yes

## Data Availability

The data presented in this study are available on request from the corresponding author.

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
