# Peer review of "OpenBloodFlow: A User-Friendly OpenCV-Based Software Package for Blood Flow Velocity and Blood Cell Count Measurement for Fish Embryos"

_biology, 2022, doi:10.3390/biology11101471_

Round 1

Reviewer 1 Report

The study purposed the use of the new OpenCV software that developed by the authors to measure the blood flow velocity and cell count. They compared their OpenCV with other  established methods and claimed that their software has certain advantages among others. The work is well written, and easy to follow. Methodology is clearly descripted. I have several comments for authors to improve the work:

Regarding the limitation, from L454, are these limitations also occur in other methods? Please explain.

Authors used the wild-type zebrafish in their assay. I wonder if the common GFP line (like fli1a or others) images will affect the accuracy of the result?

fig 7 legend, L402, "." Zebrafish

Author Response

Comments and Suggestions for Authors

The study purposed the use of the new OpenCV software that developed by the authors to measure the blood flow velocity and cell count. They compared their OpenCV with other established methods and claimed that their software has certain advantages among others. The work is well written, and easy to follow. Methodology is clearly descripted. I have several comments for authors to improve the work:

Regarding the limitation, from L454, are these limitations also occur in other methods? Please explain.

Thank you for the concern regarding the limitation of the current and previous method. In this study the authors provide the comparison with other method using trackmate Plugin in ImageJ software which is previously done by Santoso et al. Regarding the limitation from the previous method, because ImageJ ROI selection was done manually, so there should be no mistakes regarding the ROI selection. However, it should be noted that ImageJ method also based on 2D videography that also depends on the quality and the clarity of the video. Thus, some reduction in the accuracy should be expected in ImageJ method if the quality of the recorded video was not good enough to provide the exact location of the blood cell. Recent study by Ye et al. also mention that the limitation of their study was the lack of resolution that limit the precision of their tool.  Thus, the authors agree that the lack of quality and clarity in the video will be the limiting factor of every tools that depends on the 2D videography.

Santoso, F.; Sampurna, B.P.; Lai, Y.-H.; Liang, S.-T.; Hao, E.; Chen, J.-R.; Hsiao, C.-D. Development of a simple imagej-based method for dynamic blood flow tracking in zebrafish embryos and its application in drug toxicity evaluation. Inventions 2019, 4, 65.

Maung Ye, S.S.; Kim, J.K.; Carretero, N.T.; Phng, L.-K. High-Throughput Imaging of Blood Flow Reveals Developmental Changes in Distribution Patterns of Hemodynamic Quantities in Developing Zebrafish. Frontiers in physiology 2022, 1150.

Authors used the wild-type zebrafish in their assay. I wonder if the common GFP line (like fli1a or others) images will affect the accuracy of the result?

Thank you for the concern regarding the versatility of the tool. The most significant advantage of our study is no need to use fluorescent transgenic lines to highlight the blood cells. The blood flow rate and blood flow counts can be directly extracted from regular high-speed videography using regular wild type strains, like AB line. If using GFP line to do the same job, it will need to use high-speed detection head in laser scan confocal microscope and that will be extremely expensive and will be unaffordable to most of small research lab.  Although the video contrast was based on greyscale pixels value, in case of fluorescent images, the algorithm would only compute the displacement of moving blood cells instead of color reflection. The results of blood flow velocity would not affect, but blood cells count may get affect due to the masking on contrast region.

fig 7 legend, L402, "." Zebrafish

Thank you for the thorough review. The mistyping has been revised in the updated manuscript.

Reviewer 2 Report

Manuscript ID: biology-1925943

Title: OpenBloodFlow: A User-Friendly OpenCV-based Software Package for Blood Flow Velocity and Blood Cell Count Measurement for Fish Embryos.

Summary:

In this paper, the authors have reiterated an importance of cardiovascular biomarkers for the assessment of toxicology and pharmacology. One of the important biomarkers of this field is Blood flow including the blood count. Well, there are several methods available, however, most of them rely on paid software. Here, authors have employed a free software called OpenBloodFlow and used it to monitor blood flow and precise counting of blood cells under control and treated conditions. The team has used two popular animal models viz, embryos of zebrafish and medaka fish. They used phenylhydrazine or ractopamine to experimentally induce alteration in blood flow in these embryos. The software “OpenBloodFlow” is a python-based software based on OpenCV.

Comments to Authors:

Minor Comments:

1. On page 3, Table 1, Parker et al. (2014) (row), Measurement Principle (column), authors mentioned "Detection of pixel changes in the density of cardiac muscles...". Instead, they can write "Detection of change in pixel density of cardiac muscles...".

This is because in the paper Parker et al, has mentioned on page 3 of their publication that “……This software detects changes in pixel density associated with cardiac muscle contraction and chamber filling, and registers this as contractions of the cardiac muscle in beats per minute (bpm)…”.

2. On page 5, line 108, please mention the final DMSO percentage in each treatment group. As the final concentration of DMSO in treatment groups should not exceed 0.1% of DMSO.  If its more than that, authors can explain the reason or give reference to support their claim.

3. It is expected to see the control groups should have DMSO treatment. It is not clear from the text if the authors have treated the control group with blank DMSO or not?

4. On page 14, line 382-384, the authors have mentioned about different figures Figure A2A, A2B, A2C. It is not very intuitive to understand if the figures are from main sections or from Appendix. So, instead of writing Appendix, its better to write Supplementary Figure S1A, S1B, S2C etc. Please shift the appendix to supplementary file. 

5. I wonder if the authors have access to the commercially available software known for blood flow measurements? If yes, then it would be interesting to compare their OpenCV results with the commercial software too. Though it is not necessary for the revision of this paper.

Major Comments:

1. The authors have seen an interesting result of decrease in cell count at 4 dpf onwards. However, they did not discuss this result. Though, you have seen the similar decline in cell numbers even using ImageJ method, it would be interesting if the authors would also discuss the physiological cause of this decline. Does this decline correspond to increase in fish opacity?

2. One of the interesting parts of the paper is advantage of current method involving OpenCV. This section (page 16) is very lengthy and seems messy. I want the authors to please focus on this section and rework on it to highlight the major points of advantages this method offers to users over other available methods. May be the authors can use a Table to describe the various advantages the present method offers along with the limitations of other available methods in the same table. This will increase the readability of the text and the manuscript as a whole.

3. In this manuscript, the authors have built their point for method using OpenCV based on its cost-effectiveness. As their method relied on the use of Fast CCD camera (up to 200 fps speed), what would be the cost factor of this camera in this method compared to the complete system for blood flow measurement available commercially?

Author Response

Comments to Authors:

Minor Comments:

  1. On page 3, Table 1, Parker et al. (2014) (row), Measurement Principle (column), authors mentioned "Detection of pixel changes in the density of cardiac muscles...". Instead, they can write "Detection of change in pixel density of cardiac muscles...".

This is because in the paper Parker et al, has mentioned on page 3 of their publication that “……This software detects changes in pixel density associated with cardiac muscle contraction and chamber filling, and registers this as contractions of the cardiac muscle in beats per minute (bpm)…”.

Thank you for the suggestion. As the reviewer mentioned, the word choice in table 1 could be confusing to the reader. Thus, the word has been rephrased in the updated manuscript according to the reviewer suggestion.

  1. On page 5, line 108, please mention the final DMSO percentage in each treatment group. As the final concentration of DMSO in treatment groups should not exceed 0.1% of DMSO.  If its more than that, authors can explain the reason or give reference to support their claim.

Thank you for the concern regarding the usage of DMSO as a solvent. As already mentioned in manuscript, both compounds were made into 1000 ppm stock solution and further diluted to 0.15 ppm of PHZ and 4 ppm of RAC at the time of exposure. Thus, in this case, the control mentioned in the manuscript was refer to solvent control which is 0.05% of DMSO in ddH2O. However after calculation, the final DMSO concentration used was 0.015% in PHZ and 0.4% in RAC. Some concern regarding the high concentration of DMSO has been studied recently and several studies suggest that DMSO concentration up to 1% has little to no effect into zebrafish larvae (Hallare et al. 2006, Hoyberghs et al. 2001). The authors agree that addition of explanation regarding this concern will benefit the manuscript. Thus, the author already updated the manuscript according to reviewer suggestion.

Hallare, A.; Nagel, K.; Köhler, H.-R.; Triebskorn, R. Comparative embryotoxicity and proteotoxicity of three carrier solvents to zebrafish (Danio rerio) embryos. Ecotoxicology and environmental safety 2006, 63, 378-388.

Hoyberghs, J.; Bars, C.; Ayuso, M.; Van Ginneken, C.; Foubert, K.; Van Cruchten, S. DMSO Concentrations up to 1% are Safe to be Used in the Zebrafish Embryo Developmental Toxicity Assay. Frontiers in Toxicology 2021, 3.

  1. It is expected to see the control groups should have DMSO treatment. It is not clear from the text if the authors have treated the control group with blank DMSO or not?

Thank you for the comment. As the mentioned above. This study used the solvent control which is 0.05% DMSO as the control. The authors also agree with the reviewer that this should be mentioned in the manuscript. Thus, the detail of the control used has been added in the revised manuscript.

  1. On page 14, line 382-384, the authors have mentioned about different figures Figure A2A, A2B, A2C. It is not very intuitive to understand if the figures are from main sections or from Appendix. So, instead of writing Appendix, its better to write Supplementary Figure S1A, S1B, S2C etc. Please shift the appendix to supplementary file. 

Thank you for the wonderful suggestion. The author also agree that the writing of Figure A2A will confuse the reader. Thus, the writing style for all the supplementary figure has been changed to accommodate this problem in the revised manuscript.

  1. I wonder if the authors have access to the commercially available software known for blood flow measurements? If yes, then it would be interesting to compare their OpenCV results with the commercial software too. Though it is not necessary for the revision of this paper.

Thank you for the comment. As per manuscript making, the authors did not have the access to the commercially available software for blood flow measurement. Thus, doing comparison with another tool other than freeware ImageJ software will be difficult to perform at this moment. However, the authors also agree that comparing the result of commercially available tool would be an interesting topic for future study to standardize the blood flow measurement method in zebrafish.

Major Comments:

  1. The authors have seen an interesting result of decrease in cell count at 4 dpf onwards. However, they did not discuss this result. Though, you have seen the similar decline in cell numbers even using ImageJ method, it would be interesting if the authors would also discuss the physiological cause of this decline. Does this decline correspond to increase in fish opacity?

Thank you for the comment. Regarding the decreased cell count on the fourth day onwards, due to the very few study about this matter prior to the current study, the authors cannot find exact reason to this problem. However, the authors speculated that the reduction of blood cell count was due to the differentiation of blood cells that happened during the hematopoiesis that happen during the growth that change the shape of the blood cell and might change the number of blood composition (Gore et al., 2018). The authors agree that the physiological cause could be briefly discussed to increase the manuscript quality. Thus, the manuscript has been updated according to reviewer suggestion.

Gore, A. V., Pillay, L. M., Venero Galanternik, M., & Weinstein, B. M. (2018). The zebrafish: A fintastic model for hematopoietic development and disease. Wiley Interdisciplinary Reviews: Developmental Biology, 7(3), e312.

  1. One of the interesting parts of the paper is advantage of current method involving OpenCV. This section (page 16) is very lengthy and seems messy. I want the authors to please focus on this section and rework on it to highlight the major points of advantages this method offers to users over other available methods. May be the authors can use a Table to describe the various advantages the present method offers along with the limitations of other available methods in the same table. This will increase the readability of the text and the manuscript as a whole.

Thank you for the wonderful suggestion. The author agree that the section mentioned above was too long and need to rewrite to increase the reviewer comprehension about the manuscript. Thus, additional table has been added in the updated manuscript and the section has been compressed heavily to only point up the highlight of the paragraph according to the reviewer suggestion.

  1. In this manuscript, the authors have built their point for method using OpenCV based on its cost-effectiveness. As their method relied on the use of Fast CCD camera (up to 200 fps speed), what would be the cost factor of this camera in this method compared to the complete system for blood flow measurement available commercially?

Thank you for the comment. In this study, the authors used high-speed CCD camera, which mounted to inverted microscope to record the video of zebrafish dorsal aorta region. Compared to other method, which already mentioned in table 1 of the manuscript, this videotaping setting was the most cost-effective setting as no specialized tools was needed to get the image of blood cell in dorsal aorta region. Furthermore, the downstream analysis was done using OpenCV module from Python software, which is a free software. Thus in term of cost, the expense to do the analysis using this setting was only come from the CCD camera and inverted microscope which is consider cheaper than specialized tool.  This high-speed CCD is made by Taiwan company and the total price is only around 1/5 comparing to other commercial products (like third-party tools from MicroZebraLab™ or DanioScope™).